# Plants capable of selfing are more likely to become naturalized

Mialy Razanajatovo[1], Noëlie Maurel[1], Wayne Dawson[2], Franz Essl[3,4], Holger Kreft[5], Jan Pergl[6], Petr Pyšek[6,7], Patrick Weigelt[5], Marten Winter[8] & Mark van Kleunen[1]

Many plant species have established self-sustaining populations outside their natural range because of human activities. Plants with selfing ability should be more likely to establish outside their historical range because they can reproduce from a single individual when mates or pollinators are not available. Here, we compile a global breeding-system database of 1,752 angiosperm species and use phylogenetic generalized linear models and path analyses to test relationships between selfing ability, life history, native range size and global naturalization status. Selfing ability is associated with annual or biennial life history and a large native range, which both positively correlate with the probability of naturalization. Path analysis suggests that a high selfing ability directly increases the number of regions where a species is naturalized. Our results provide robust evidence across flowering plants at the global scale that high selfing ability fosters alien plant naturalization both directly and indirectly.

[1] Ecology, Department of Biology, University of Konstanz, Universitätsstrasse 10, Konstanz D-78457, Germany. [2] Conservation Ecology Group, Department of Biosciences, Durham University, South Road, Durham DH1 3LE, UK. [3] Division of Conservation, Vegetation and Landscape Ecology, University of Vienna, Wien 1030, Austria. [4] Centre for Invasion Biology, Department of Botany and Zoology, Stellenbosch University, Matieland 7602, South Africa. [5] Biodiversity, Macroecology and Biogeography, University of Göttingen, Büsgenweg 1, Göttingen D-37077, Germany. [6] Institute of Botany, Department of Invasion Ecology, The Czech Academy of Sciences, Průhonice, CZ-25243, Czech Republic. [7] Department of Ecology, Faculty of Science, Charles University, Viničná 7, Prague 2, CZ-12844, Czech Republic. [8] German Centre for Integrative Biodiversity Research (iDiv) Halle-Jena-Leipzig, Deutscher Platz 5e, Leipzig D-04103, Germany. Correspondence and requests for materials should be addressed to M.R. (email: mialy.razanajatovo@uni-konstanz.de).

The human-caused introduction and subsequent spread of species into new regions has become a defining feature of global environmental change in the Anthropocene[1]. At least 3.9% of species in the global flora has now established self-sustaining populations in regions where they did not occur naturally[2]. Most such naturalized plants have been introduced for horticulture[3], and subsequently escaped from gardens, nurseries and arboreta[4]. Other plant naturalizations have resulted from accidental introductions, for example as seed contaminant of imported crops[5]. Some naturalized plants have become invasive in many regions of the world, where they often have detrimental impacts on the environment and human societies[6]. Naturalization is a key stage of the invasion process when species overcome the barriers that prevent them from establishing self-sustaining populations in the wild[7,8]. Thus, understanding what drives naturalization is a key biological question of global interest[9].

The ability to reproduce is of particular importance for understanding naturalization, because propagule supply is required for the founding and maintenance of populations[10,11]. Baker[12] posed that species capable of uniparental reproduction are more likely to establish after long-distance dispersal, because they can reproduce from a single individual. This hypothesis, known as Baker's Law or Baker's Rule[13], may apply to natural long-distance dispersal events as well as to species introduced by humans to new regions, because suitable mates and/or pollinators may be scarce there[14]. When mates are scarce, self-compatible species should have an advantage over self-incompatible ones. Furthermore, when pollinators are scarce, those self-compatible species that can self-pollinate—and thus are autofertile—should especially have a major advantage. The role of the breeding system could thus be most relevant during naturalization[14].

Previous tests of Baker's Law with alien plants have yielded contradicting findings. While many naturalized and invasive plants have high selfing ability in their alien range[15–18], some globally invasive plants do not possess this ability[19,20]. Some studies even suggest that the breeding system only plays a minor role in naturalization[21]. However, previous works were limited to specific taxonomic or functional groups of plants, to the invasion rather than the naturalization stage and/or to a restricted part of the globe. Thus, we still lack a comprehensive test of the role of breeding systems in explaining naturalization of alien plant species at the global scale.

A species' breeding system has commonly been defined as a qualitative trait[22,23] (for example, self-compatible versus self-incompatible). However, species classified as self-compatible might not be fully self-compatible, and those classified as self-incompatible might be partly self-compatible[24]. Furthermore, as shown in some species, even self-compatible ones, selfing may not occur at all while in others selfing may occur rarely, sometimes, often or nearly always[22]. This continuity in the degree of self-compatibility (SC) and autofertility (AF) may be important for naturalization, and thus, quantitative metrics of selfing ability are preferable. Despite the extensive number of studies on the breeding systems of flowering plants around the globe, only few efforts have been made to compile the individual quantitative findings into a global database[22,25]. Such an approach, covering as many regions and plant species as possible, is necessary to address general scientific questions related to plant reproductive strategies.

Several plant characteristics are often reported as being associated with naturalization as well as with selfing ability. First, short-lived species such as annuals and biennials predominantly occur in habitats subjected to frequent disturbances, and therefore often colonize anthropogenic habitats (that is, they become weedy[26]), and are also often selfers[14,27,28]. Second, species with selfing ability are more likely to have larger native

geographic ranges than those lacking this ability[29,30], and species with large geographic ranges or a wide habitat breadth are more likely to naturalize elsewhere[31–33]. It is thus important to unravel the complex causal associations among these different factors to assess whether relationships between selfing ability and naturalization, if any, are direct or indirect. Furthermore, species with different degrees of evolutionary relatedness might have different degrees of similarity in plant traits associated with naturalization and in traits associated with selfing ability. Therefore, it is necessary to account for other traits and phylogeny[34] when testing the relationship between naturalization and breeding system.

To test the effects of SC and AF on naturalization at the global scale, we assembled a global breeding-system database of flowering plants. We quantified species' selfing ability using indices of SC and AF[22,35]. We combined these data with information on life history, native range size and global information on the incidence and the extent of naturalization (expressed as being naturalized or not outside the native range and the number of regions where naturalized, respectively) using the Global Naturalized Alien Flora database (GloNAF)[2], which is the most comprehensive data source on naturalized alien plant species distributions. We used a phylogenetically informed generalized linear model approach to account for the effect of relatedness among species. Further, we used path analysis to separate the potential direct effects of SC and AF on naturalization from the indirect associations caused by correlations with life history and native range size. Our questions were: (i) are the incidence (whether or not a species has naturalized somewhere) and extent of naturalization (the number of regions where naturalized) of alien plant species associated with SC and AF, as suggested by Baker's Law? (ii) On the basis of a path analysis, what is the strength of potential direct and indirect effects of SC and AF on the incidence and extent of naturalization when accounting for life history and native range size?

Here we show that SC and AF are positively related to whether a species has naturalized somewhere and the number of regions where a species is naturalized, both directly and indirectly. Furthermore, path analysis suggests that SC and AF are associated with annual or biennial life history and a large native range, which both positively correlate with the probability of naturalization. High SC and AF also increase with the number of regions where a species is naturalized. Our results provide evidence of the validity of Baker's Law across flowering plants at the global scale, and of its significance for alien plant naturalization.

## Results

**Global patterns in self-compatibility and autofertility.** Based on a comprehensive literature search, we found breeding-system data for a total of 1,752 species (Supplementary Data 1) covering all major clades of angiosperms (Supplementary Fig. 1), from 161 families (Supplementary Fig. 2), and from all over the world (Supplementary Fig. 3). The indices of SC and AF had a continuous bimodal frequency distribution from zero (indicating absence of selfing ability) to one (indicating high selfing ability), and were higher for short-lived (annual and biennial) species than for perennials (Fig. 1).

**Naturalization incidence associated with selfing ability.** Of the 1,752 species in our database, 498 have become naturalized somewhere. Although the naturalization incidence appeared to be positively associated with SC and AF indices, calculated for fruit set, that is, fruit/flower ratio (Fig. 2a,b), these effects were not significant in the phylogenetic logistic regressions (Table 1).

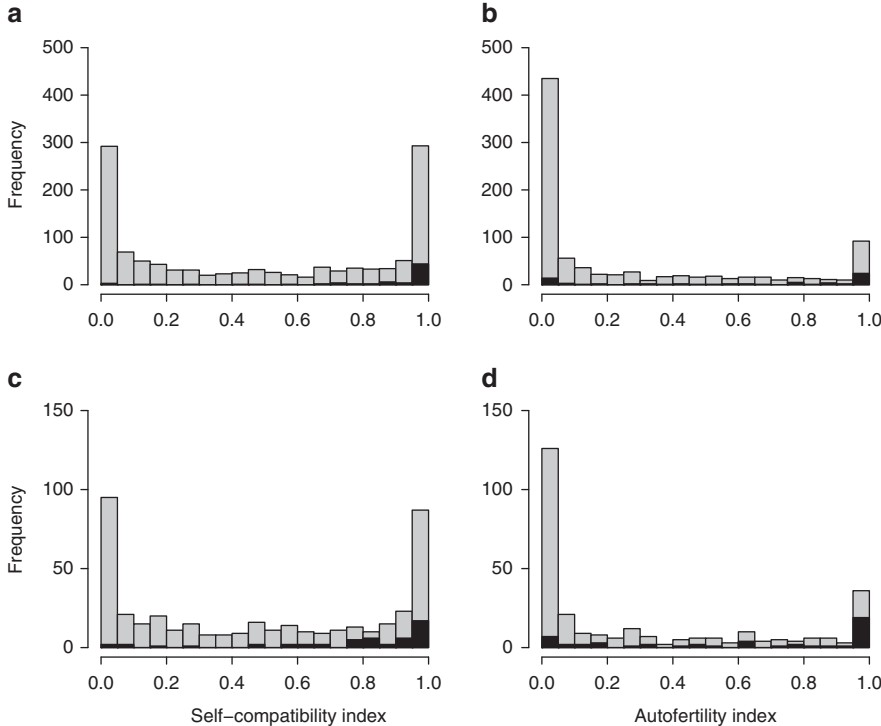

**Figure 1 | Global patterns in self-compatability and autofertility.** Frequency distributions of four indices of selfing ability compiled in our global database on breeding systems of angiosperms. (**a**) SC index calculated based on fruit set ($n = 1,192$). (**b**) AF index calculated based on fruit set ($n = 873$). (**c**) SC index calculated based on the number of seeds per flower ($n = 422$). (**d**) AF index calculated based on the number of seeds per flower ($n = 286$). Sample sizes refer to the total number of species from individual studies in the breeding-system database. Annual and biennial species are shown in black and perennial ones in grey.

However, a species was significantly more likely to naturalize somewhere if its native range was large, as an increase of one standard deviation (0.854 in the analysis with SC, and 0.848 in the analysis with AF) in the log-transformed number of regions where a species is native increased its odds of being naturalized somewhere by a factor of $e^{0.710} = 2.034$ and $e^{0.720} = 2.054$ (Table 1). Moreover, annuals and biennials had a higher naturalization incidence than perennials, as an annual or a biennial life history increased a species' odds of being naturalized somewhere by a factor of $e^{0.731} = 2.077$ compared with a perennial life history, at least so in the analysis with AF (Table 1). Although the logistic regressions that accounted for phylogeny had a better fit than the ones without phylogeny, the results of the latter were qualitatively similar (Supplementary Table 1). Path analysis, in which we could not account for phylogeny, but which allowed us to explicitly differentiate between direct and indirect relationships, confirmed that the degrees of SC and AF of species had no significant direct effects on naturalization incidence (Fig. 3a,b). However, species with a high SC and AF were more likely to naturalize, as they were likely to have a large native range, which in turn can be inferred to increase their ability to naturalize (Fig. 3a,b). Moreover, SC and AF were highest for species with an annual or biennial life history, which were more likely to naturalize than perennials (Fig. 3a,b). Although the results were slightly different for SC and AF indices based on seed production per flower instead of fruit set, these differences were small. Then, naturalization incidence was also not significantly correlated with SC and AF in the phylogenetic logistic regressions (Table 1; Fig. 2e,f). However, in the path analysis, the indirect relationship between SC, native range size and naturalization incidence was significant (Fig. 3e), but such an indirect relationship was not significant in the analysis with AF (Fig. 3f).

**Naturalization extent associated with selfing ability.** Among the 498 species in the breeding-system database that have become naturalized somewhere, the naturalization extent (measured as the number of regions) ranged from 1 to 409 regions (out of 843) of the GloNAF database. In the phylogenetic linear regression, the naturalization extent was significantly positively related to the SC and AF indices, calculated for fruit set (Table 1 and Fig. 2c,d). An increase of 1 s.d. in the SC index (0.410) and AF index (0.374) of a species increased the log-transformed number of regions where it is naturalized by 0.256 and 0.472, respectively (Table 1). The extent of naturalization was not significantly correlated with native range size, but annuals and biennials had a larger naturalization extent than perennials (Table 1). An annual or a biennial life history increased the log-transformed number of regions where a species is naturalized by 1.463 and 1.048 compared with a perennial life history in the analysis of SC and AF, respectively (Table 1). The results were qualitatively similar when not correcting for phylogeny (Supplementary Table 1). Path analysis confirmed that the degrees of SC and AF of species as well as life history had significant direct relationships with naturalization extent (Fig. 3c,d). An increase of one standard deviation in the SC index (0.410) and AF index (0.374) of a species increased the log-transformed number of regions where it is naturalized by 0.128 and 0.244 s.d., respectively. As SC and AF were highest for species with an annual or biennial life history, they were also indirectly correlated with naturalization extent (Fig. 3c,d). The paths from selfing ability to native range size and from native range size to naturalization extent were not significant (Fig. 3c,d). Although the results were slightly different for SC indices based on seed production per flower instead of fruit set, these differences were small. Then, the correlation between SC and naturalization extent was marginally significant (Table 1). However, in the path analysis, the degree of SC had a

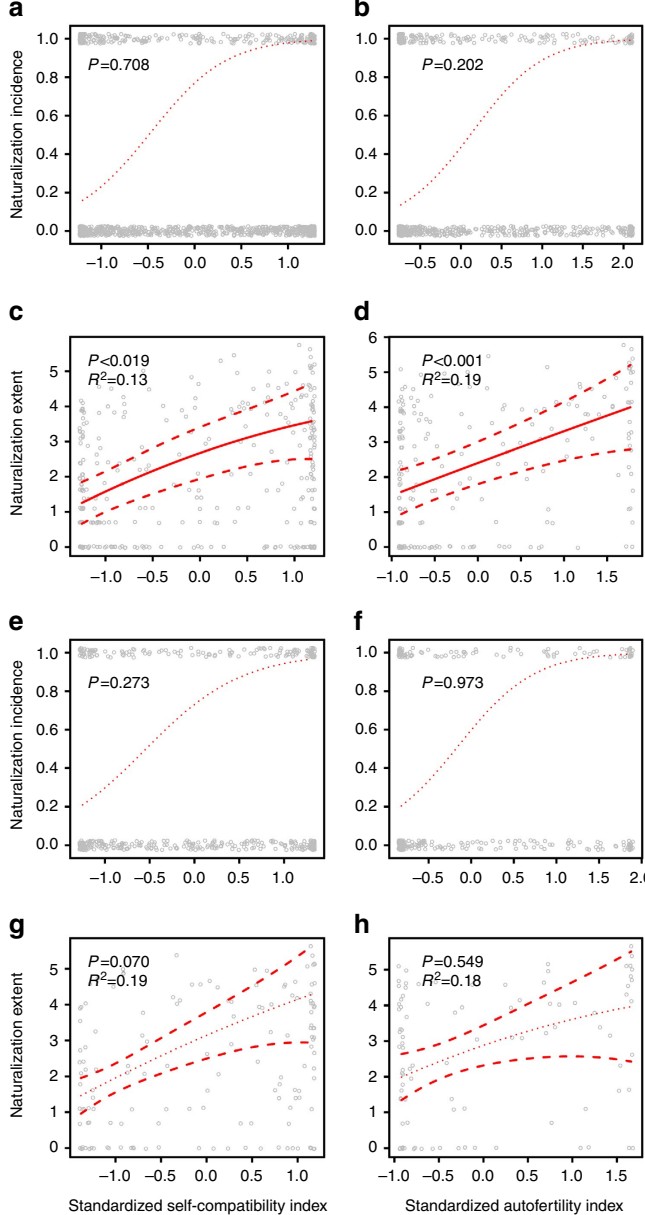

**Figure 2 | Global naturalization of alien plant species in relation to selfing ability.** (**a–d**) Analysis with indices calculated using fruit set. (**a**) A phylogenetic logistic regression testing how naturalization incidence (expressed as being naturalized somewhere or not) depends on SC index ($n = 1,181$). (**b**) A phylogenetic logistic regression testing how naturalization incidence depends on AF index ($n = 866$). (**c**) A phylogenetic linear model testing how naturalization extent (natural log-transformed number of regions where the species is naturalized) depends on SC index ($n = 295$). (**d**) A phylogenetic linear model testing how naturalization extent depends on AF index ($n = 203$). (**e–h**) Analysis with indices calculated using seed production per flower. (**e**) A phylogenetic logistic regression testing how naturalization incidence depends on SC index ($n = 417$). (**f**) A phylogenetic logistic regression testing how naturalization incidence depends on AF index ($n = 282$). (**g**) A phylogenetic linear model testing how naturalization extent depends on SC index ($n = 144$). (**h**) A phylogenetic linear model testing how naturalization extent depends on AF index ($n = 95$). Selfing-ability indices were standardized to a mean of zero and an s.d. of 1. Sample sizes refer to the total number of species from individual studies in the breeding-system database. Solid lines and dotted lines indicate significant and nonsignificant relationships, respectively. Dashed lines represent standard errors of the phylogenetic linear regression. Denoted $R^2$ are total explained variance by the variables in the model.

significant direct relationship with the naturalization extent (Fig. 3g). The results were different for AF indices based on seed production per flower as then the effect of AF on naturalization extent was not significant (Table 1 and Figs 2h and 3h).

## Discussion

We combined the largest quantitative database on breeding systems of angiosperms collected to date with the most comprehensive database on alien plant species distributions (GloNAF[2]) to assess potential relationships between selfing ability and naturalization. Our models explained 13–24% of the variation in naturalization incidence and extent of species (Table 1 and Figs 2 and 3), which is comparable with the explanatory power of models in previous ecological studies testing the importance of different traits on the naturalization or invasion of alien species (for example, ref. 36). We showed that both incidence and extent of naturalization were positively associated with SC and AF. However, the relationship between selfing ability and whether or not a species has naturalized somewhere was only indirect through a correlation with life history, and a positive relationship with native range size. In contrast, selfing ability was positively associated with the number of regions where a species has naturalized, and selfing ability significantly accounted for 36–59% of the explained variation in naturalization relative to the other variables in the models (Table 1). Similar direct effects have been found at local to regional scales. For example, many South African Iridaceae species that have become naturalized elsewhere have, in the native range, a higher ability for autonomous selfing than their non-naturalized counterparts[37]. Many successful naturalized alien plants also demonstrated high selfing ability in the introduced range[16,18,33]. Our results show for the first time that the association between selfing ability and naturalization holds at a global scale across many families.

SC provides plants with the ability to reproduce even when their populations are small—as is usually the case during the initial establishment—and mating is likely to involve few and closely related individuals (Baker's Law[12,14]). Indeed, SC confers a fitness advantage when mates are limited (reproductive assurance[28]). Selfing may increase seed production[38] and therefore propagule pressure, which is one of the major drivers of naturalization and invasion[39]. Modes of self-pollination are diverse[22,40]; some self-compatible plants still require pollinators whereas others can do autonomous self-pollination, resulting in AF. Geitonogamy, that is, pollen transfer from a flower to another flower of the same plant, and also facilitated autogamy only occur when the introduced plants encounter suitable pollinators in the new environment. However, when pollinators are limited, AF is required to assure reproduction[40,41]. As the indices of SC and AF are not independent, and both are associated with naturalization, it is not possible to infer from our study whether selfing ability is more important for overcoming mate or pollinator limitation. Recently, Pannell et al.[14] proposed a narrower circumscription of Baker's Law focusing on the consequences of mate rather than pollinator limitation. However, in a recent study, it was shown that naturalized species might attract more floral visitors than cultivated, but non-naturalized alien species[42], suggesting that pollinator limitation may be important and should not be ignored.

While our study suggests that high selfing ability may contribute to the naturalization of alien plant species, questions regarding the consequences of selfing for the demography of those species may arise. A major disadvantage of selfing is inbreeding depression[43]. Although many naturalized plants in our database have a high degree of selfing ability, some have also been shown to have substantial inbreeding depression, for

**Table 1 | Global naturalization of alien plant species in relation to selfing ability.**

| Response variables | Naturalization incidence* | | | | Naturalization extent† | | | | |
|---|---|---|---|---|---|---|---|---|---|
| **Explanatory variables** | **Estimates** | **s.e.** | **z** | **P** | **Estimates** | **s.e.** | **t** | **P** | **R²‡** |
| *Selfing-ability indices calculated using fruit set* | | | | | | | | | |
| Analysis with self-compatibility | | $n = 1{,}181$ | | | | $n = 295$ | | | 0.13 |
| Intercept | − 1.146 | 0.126 | − 9.07 | <0.0001 | 1.924 | 0.264 | 7.28 | <0.0001 | — |
| Native range size§ | 0.710 | 0.082 | 8.67 | <0.0001 | 0.101 | 0.096 | 1.05 | 0.293 | 7.14 |
| Annual/biennial§ | 0.568 | 0.418 | 1.36 | 0.174 | 1.463 | 0.475 | 3.08 | 0.002 | 50.00 |
| Self-compatibility index§ | 0.026 | 0.070 | 0.37 | 0.708 | 0.256 | 0.108 | 2.36 | 0.019 | 35.71 |
| Annual/biennial§, × self-compatibility§ | 0.243 | 0.362 | 0.67 | 0.502 | − 0.373 | 0.444 | − 0.84 | 0.402 | 7.14 |
| Analysis with autofertility | | $n = 866$ | | | | $n = 203$ | | | 0.19 |
| Intercept | − 1.276 | 0.171 | − 7.47 | <0.0001 | 2.241 | 0.294 | 7.69 | <0.0001 | — |
| Native range size§ | 0.720 | 0.097 | 7.42 | <0.0001 | 0.033 | 0.120 | 0.27 | 0.785 | 0.10 |
| Annual/biennial§ | 0.731 | 0.352 | 2.08 | 0.038 | 1.048 | 0.387 | 2.70 | 0.007 | 40.46 |
| Autofertility§ | 0.104 | 0.081 | 1.28 | 0.202 | 0.472 | 0.132 | 3.58 | <0.001 | 59.15 |
| Annual/biennial§, × autofertility§ | 0.319 | 0.241 | 1.32 | 0.186 | − 0.030 | 0.302 | − 0.10 | 0.920 | 0.05 |
| *Selfing-ability indices calculated using seed production per flower* | | | | | | | | | |
| Analysis with self-compatibility | | $n = 417$ | | | | $n = 144$ | | | 0.19 |
| Intercept | − 0.660 | 0.152 | − 4.35 | <0.0001 | 2.327 | 0.147 | 15.85 | <0.0001 | — |
| Native range size§ | 0.618 | 0.126 | 4.89 | <0.0001 | 0.155 | 0.148 | 1.05 | 0.296 | 6.44 |
| Annual/biennial§ | 0.502 | 0.42 | 1.19 | 0.233 | 1.480 | 0.455 | 3.25 | 0.001 | 66.48 |
| Self-compatibility index§ | 0.121 | 0.110 | 1.09 | 0.273 | 0.275 | 0.150 | 1.82 | 0.070 | 25.62 |
| Annual/biennial§, × self-compatibility§ | − 0.058 | 0.381 | − 0.15 | 0.880 | − 0.237 | 0.475 | − 0.50 | 0.619 | 1.46 |
| Analysis with autofertility | | $n = 282$ | | | | $n = 95$ | | | 0.18 |
| Intercept | − 0.688 | 0.186 | − 3.69 | <0.001 | 2.707 | 0.250 | 10.81 | <0.0001 | — |
| Native range size§ | 0.900 | 0.176 | 5.11 | <0.0001 | 0.068 | 0.189 | 0.36 | 0.720 | 1.10 |
| Annual/biennial§ | 0.298 | 0.407 | 0.72 | 0.473 | 1.624 | 0.488 | 3.33 | 0.001 | 86.94 |
| Autofertility§ | 0.004 | 0.145 | 0.03 | 0.973 | 0.121 | 0.201 | 0.60 | 0.549 | 7.04 |
| Annual/biennial§, × autofertility§ | 0.276 | 0.316 | 0.87 | 0.382 | − 0.311 | 0.410 | − 0.76 | 0.449 | 4.91 |

Results of four phylogenetic logistic regressions and four phylogenetic linear regressions testing the global naturalization of alien plant species in relation to selfing ability (measured as self-compatibility and autofertility indices), native range size, annual/biennial versus perennial life history and the interaction between life history and selfing ability. Global naturalization was measured as naturalization incidence outside the native range (expressed as being naturalized somewhere or not) and naturalization extent (natural log-transformed number of regions where the species is naturalized). Native range size was measured as the number of TDWG level-2 regions. Self-compatibility and autofertility indices were calculated using fruit set and seed production per flower from our global breeding-system database. Sample sizes refer to the total number of species from individual studies in the breeding-system database.
*Phylogenetic logistic regression: $\alpha = 0.035$ in the analysis with self-compatibility and 0.026 in the analysis with autofertility (indices calculated using fruit set); and $\alpha = 0.033$ in the analysis with self-compatibility and 0.029 in the analysis with autofertility (indices calculated using seed production per flower).
†Phylogenetic linear model: $\lambda = 0.141$ in the analysis with self-compatibility and 0.142 in the analysis with autofertility (indices calculated using fruit set); and $\lambda < 0.0001$ in the analysis with self-compatibility and 0.048 in the analysis with autofertility (indices calculated using seed production per flower).
‡Total explained variance by the variables in the model, and the relative importance of each variable calculated as the difference in deviance between the full model and a model without the variable of interest. $R^2$ could not be calculated for the phylogenetic logistic regressions.
§Variables were rescaled to have a mean of zero and an s.d. of 1.

example the tree *Acacia dealbata* Link in the introduced range in South Africa[44]. On the other hand, some self-compatible and autofertile alien plants have negligible inbreeding depression, such as the geophyte *Lilium formosanum* Wallace, also naturalized in South Africa[45]. These two examples suggest that inbreeding depression costs vary among species. Furthermore, selfing ability can vary within species (for example, in different locations[46]). The roles of intraspecific variation in selfing ability and potential short- and long-term disadvantages of selfing for naturalized alien plants require further tests.

Whether a plant species has naturalized outside the native range and how widely it has naturalized might depend on characteristics of the mode of introduction such as introduction effort or pathway, and the spatial distribution of propagules[47]. The species in our study might not have the same probability of being introduced to all potentially suitable regions outside their native ranges[48,49]. Particularly for species that have not become naturalized at all, one cannot exclude that they have not naturalized simply because they were not introduced. This could have obscured a direct effect of selfing ability on naturalization incidence. Nevertheless, because we accounted for native range size, which is a good predictor of the probability of being introduced elsewhere[50], variation in introduction effort is at least partly accounted for.

Species with higher selfing ability in our database cover larger native ranges (Fig. 3a,b). This finding is in line with the results of

a recent study showing that selfing plants have larger native ranges than their outcrossing close relatives[31]. These results support Baker's Law, in that the ability of plants to overcome mate limitation has contributed to a greater native range expansion[12]. Native range size itself can be a strong predictor of naturalization as shown in previous studies. For example, introduced Poaceae and Fabaceae that had larger native ranges in Asia and Africa had a larger naturalized range in North America[51]. Similarly, temperate shrubs and trees that have larger native ranges had greater naturalization success globally[29]. The probability of central European plants becoming naturalized outside their native range was also determined by native range size[52]. So, overall, species widespread at home tend to be more likely to naturalize.

There are several potential reasons why naturalization may be associated with a large native range. First, if a species is widespread and common, it is more likely to be transported and introduced elsewhere[52]. Second, a large native range could also be associated with traits conferring higher dispersal ability, such as small and numerous seeds[53] and selfing ability[31]. Third, as a large native range tends to contain a broad variety of environmental conditions, species occurring in a large geographic area may be ecologically versatile[52]. It is difficult to say which of these potential mechanisms is most important. However, the strong relationship between native range size and naturalization incidence and the absence of such relationship with naturalization

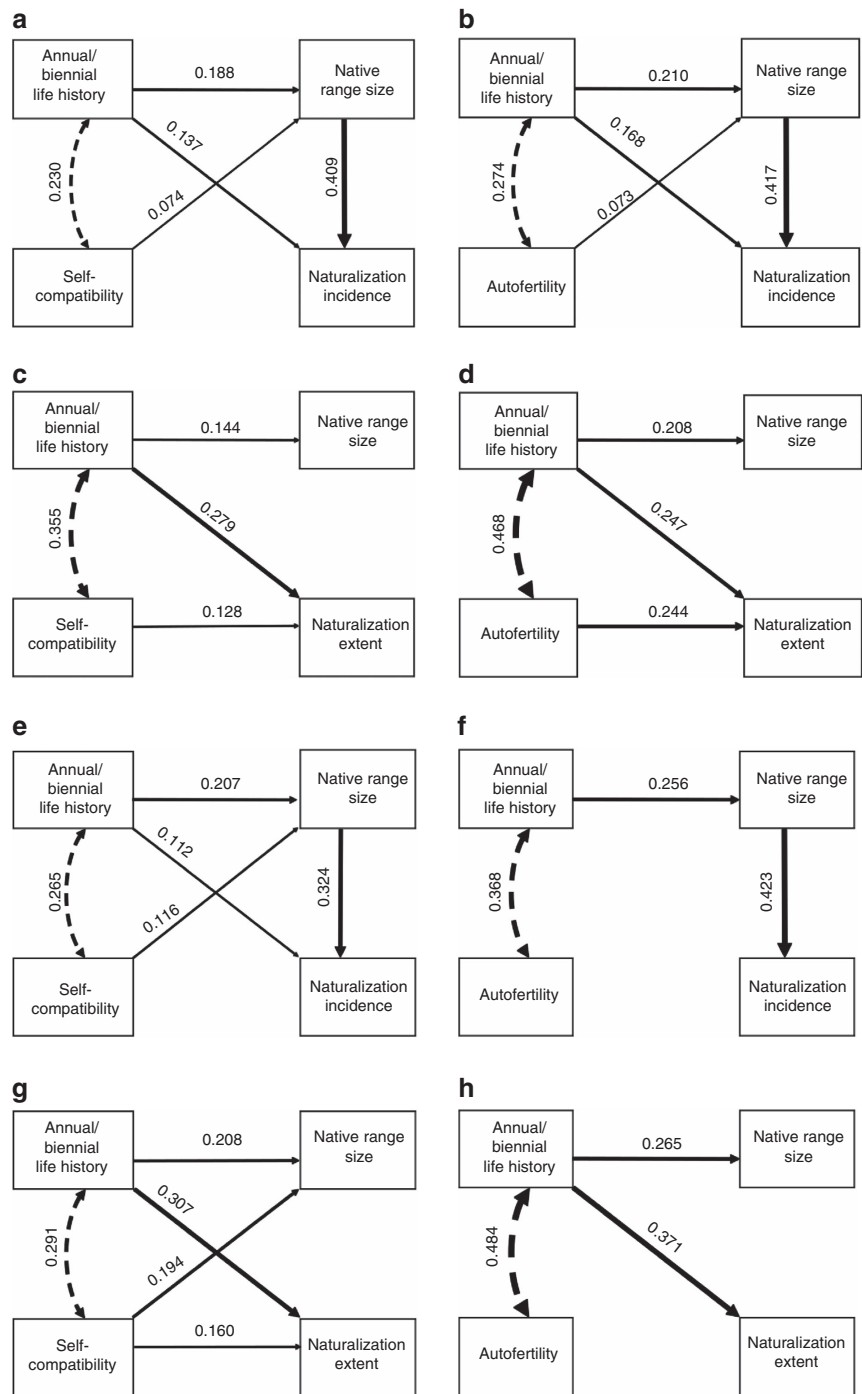

**Figure 3 | Direct and indirect association of global naturalization with selfing ability. (a–d)** Selfing-ability indices calculated using fruit set from our global breeding-system database. (**a**) Effects of SC on naturalization incidence (expressed as being naturalized somewhere or not); $n = 1,181$, no. of parameters $= 7$, CFI $= 0.993$, RMSEA $= 0.048$ and $R^2_{\text{naturalization incidence}} = 0.20$. (**b**) Effects of AF on naturalization incidence; $n = 866$, no. parameters $= 7$, CFI $= 0.988$, RMSEA $= 0.061$ and $R^2_{\text{naturalization incidence}} = 0.24$. (**c**) Effects of SC on the extent of naturalization (number of regions where the species is naturalized); $n = 295$, no. parameters $= 8$, CFI $= 1.000$, RMSEA $= 0.000$ and $R^2_{\text{naturalization extent}} = 0.19$. (**d**) Effects of AF on the extent of naturalization; $n = 203$, no. parameters $= 8$, CFI $= 1.000$, RMSEA $= 0.000$ and $R^2_{\text{naturalization extent}} = 0.21$. (**e–h**) Selfing-ability indices calculated using seed production per flower from our global breeding-system database. (**e**) Effects of SC on naturalization incidence; $n = 417$, no. parameters $= 7$, CFI $= 0.982$, RMSEA $= 0.062$ and $R^2_{\text{naturalization incidence}} = 0.14$. (**f**) Effects of AF on naturalization incidence; $n = 282$, no. parameters $= 7$, CFI $= 0.974$, RMSEA $= 0.090$ and $R^2_{\text{naturalization incidence}} = 0.22$. (**g**) Effects of SC on the extent of naturalization; $n = 144$, no. parameters $= 8$, CFI $= 0.990$, RMSEA $= 0.061$ and $R^2_{\text{naturalization extent}} = 0.18$. (**h**) Effects of AF on the extent of naturalization; $n = 95$, no. parameters $= 8$, CFI $= 0.991$, RMSEA $= 0.059$ and $R^2_{\text{naturalization extent}} = 0.18$. Sample sizes refer to the total number of species from individual studies in the breeding-system database. Curved dashed lines with two-headed arrows indicate bivariate correlations between variables; solid lines with one-headed arrow indicate directed paths connecting the variables. The thickness of each path is proportional to the value of the path coefficient. Only path coefficients that are significant (z-test, $P < 0.05$) are shown. CFI, Comparative Fit Index; RMSEA, root mean square error of approximation.

extent suggest that at least the increased likelihood of being introduced elsewhere plays a role.

Annual and biennial species were more likely to be naturalized elsewhere, and in larger numbers of regions. Many widely naturalized species are associated with habitats prone to frequent disturbances and thus have weedy characteristics in their native range, and many weedy species are annuals or short-lived plants[20]. Annual and biennial species have an advantage over perennials by reaching maturity earlier. Such characteristics of r-strategists could help alien species to rapidly colonize vacant space—due to widespread and frequent man-made disturbances—before other species do (priority effect[49]). Furthermore, annual and biennial species may depend more on reproductive assurance than perennials, as the latter have more opportunities to reproduce, and frequently can multiply vegetatively[13,14]. Thus, annuals and biennials are likely to have high selfing ability. Our results on a large set of angiosperm species suggest that, along with selfing ability, annual and biennial life histories contribute to alien plant naturalization.

We provide the first comprehensive evidence that, across a large number of angiosperms and at the global scale, selfing ability plays a significant role in the naturalization of alien plant species. Species with high selfing ability are more likely to naturalize outside their native ranges, as such species are also more likely to occupy larger native ranges, which in turn increases their ability to naturalize in new regions to which they were introduced. These results therefore provide robust evidence that high selfing ability fosters naturalization outside the native range, thereby supporting Baker's Law[12].

## Methods

**Compilation of a global database on breeding systems.** To assemble as much as possible of the published quantitative data on breeding systems of angiosperm species into a global database, we searched for plant breeding-system literature in Web of Science (http://apps.webofknowledge.com; Supplementary Fig. 4). We used the search keyword string 'TS = (('breeding system' OR 'mating system' OR 'self-compatib*' OR 'self-fertil*' OR autogam* OR 'auto-fertil*' OR outcross* OR apo-mixis) AND plant)', and searched for all indexed documents from the year 1900 onwards. The last search was done in February 2015. We scanned all 6,678 resulting titles, and excluded the ones that did not present breeding-system data for angiosperms (for example, studies on animals, cryptogams and gymnosperms), and studies that only assessed the realized outcrossing rates based on molecular markers without assessing selfing ability. Subsequently, we screened the abstracts of the remaining 1,675 articles. Studies that explicitly included all or a subset of the following treatments were selected: bagging of flowers to exclude pollinators (testing for AF), bagging of flowers in combination with self-pollination (testing for SC), bagging of flowers in combination with cross-pollination (testing for maximal fruit and seed production when bagged), bagging of flowers in combination with emasculation (testing for apomixis), supplemental hand-pollination (testing for pollen limitation) and unmanipulated control flowers (testing for fruit and seed production under natural conditions). Out of the 705 studies that fulfilled our criteria, we examined 696 full-text documents, as nine studies were not accessible. We included in our database all studies that provided measures of fruit set (fruit/flower ratio) and/or seed production (number of seeds per plant or per flower or per fruit, or seed/ovule ratio) for all or a subset of the above-mentioned treatments. Of the 696 articles, 516 yielded suitable data. We additionally included data from 244 documents that we encountered independently of the literature search (including 113 references used by Raduski et al.[25]).

For each species reported in a study, we documented all available fruit and seed production data in each of the different pollination treatments. If data were provided in the text or in tables, we extracted them directly. If the data were provided in graphs only, we extracted the data using ImageJ[54]. According to the availability of the data, we obtained for each species and treatment, the percentage of treated flowers that produced fruits (fruit set) and the number of seeds produced per flower (Supplementary Table 2). When the number of seeds per flower was not provided, we calculated it by multiplying fruit set by the number of seeds per fruit (the percentages of values that were calculated this way are given in Supplementary Table 2). We found breeding-system data for 1,829 plant taxa. Because some of these taxa may be synonyms, and to facilitate alignment of the species list with other databases (see below), we standardized the scientific names of the species in our database following The Plant List (http://www.theplantlist.org), using the package 'Taxonstand'[55] in R[56]. For 14 taxa that did not occur in The Plant List, we kept the names given in the original study from which breeding-system data were extracted.

Because a species' breeding system is often related to its life history, we included for each species, information on its life history (annual, biennial and perennial). When this was not provided in the publication from which the breeding-system data were extracted, we consulted other sources: Encyclopedia of Life (http://eol.org), Royal Botanic Gardens, Kew (http://epic.kew.org), the United States Department of Agriculture database (http://plants.usda.gov/java/), Tropicos (http://www.tropicos.org), Instituto de Botanica Darwinion (http://www.darwin.edu.ar/Proyectos/Flora Argentina/fa.htm); and did extensive internet searches.

As a broad native distribution may influence the global naturalization of an alien species, and might be affected by a plant's breeding system and life history, we compiled information on the native range size of each species in the database using the number of TDWG level-2 regions the species is native to (52 regions in total; adapted from the scheme of the Biodiversity Information Standards or Taxonomic Databases Working Group[57]). We extracted data from the World Checklist of Selected Plant Families (WCSP; http://apps.kew.org/wcsp/), the Germplasm Resources Information Network (http://www.ars-grin.gov/cgi-bin/npgs/html/index.pl), the Global Biodiversity Information Facility (http://www.gbif.org/), Tropicos (http://www.tropicos.org), the United States Department of Agriculture database (http://plants.usda.gov/java/), Encyclopedia of Life (http://eol.org), Australian National Botanic Gardens (https://www.anbg.gov.au/index.html), CIRAD (http://arbres-reunion.cirad.fr/accueil), and IUCN (http://www.iucnredlist.org/). For 21 species, we obtained information on the native range from the original study from which breeding-system data had been extracted.

To test whether global naturalization of species is related to their breeding systems, we used the Global Naturalized Alien Flora (GloNAF) database version 1.1[2], which is the most comprehensive database of naturalized plants. It includes 13,168 alien plant species that have become naturalized in a total of 843 regions (481 mainland regions and 362 island regions) on the globe (∼83% of the Earth's land area). For each species in our database, we added data on presence/absence in the GloNAF database (that is, whether or not a species is listed in GloNAF) and the number of regions where the species has become naturalized outside its native range.

We removed 30 taxa that were not identified to the species level from the breeding-system database, because we could not find information on native and naturalized ranges for these taxa. In total, after taxonomic standardization, we had quantitative breeding-system data for 1,752 angiosperm species (Supplementary Data 1) from 161 families (Supplementary Fig. 2), collected in 116 regions (covering tropics, subtropics and temperate zones, Supplementary Fig. 3) using 763 studies. Our database covers experiments performed between 1868 and 2015 in natural populations (1,380 species), plantings (61 species), common gardens (143 species) and greenhouses (269 species). The breeding-system database was not compiled with any a priori bias towards the alien species. For 127 species, the breeding system was assessed outside their native range.

**Phylogeny.** To account for potential biases due to different degrees of evolutionary relatedness among species in the analyses[34], we constructed a phylogenetic tree of the 1,752 species in our database. We first pruned an existing megatree of 31,749 plant species[58] using the 'ape' package[59] in R. The pruned tree returned 821 species in our database that matched with the tips of the megatree. We then added all species belonging to identical genera as polytomies on the pruned tree and obtained a new tree with 1,266 species. We manually added the 486 remaining missing species by supplementing nodes that represented the latest common ancestor of the added taxon. For families that did not occur in the megatree, we decided on the last common ancestor based on taxonomic information on the Angiosperm Phylogeny Website version 13 (http://www.mobot.org/MOBOT/research/APweb/). For genera that did not occur in the megatree, we created polytomies under the families. We resolved resulting polytomies using information from online and published phylogenies (Supplementary Table 3). We then obtained the topology of our phylogenetic tree, and estimated the branch length of the tree using phylocom[60] considering node ages given by Wikström[61]. For each species that was reported in more than one study (n = 125), we added new nodes leading to multiple tips in the tree. To ensure phylogenetic equality, we adjusted the within-species branch lengths to the equivalent of 100 years[62].

**Data analysis.** To quantify the degree of SC and AF of each species, we calculated an index of SC[22] and an index of AF[22,35], respectively:

$$SC = \frac{Outcome\ from\ self-pollination}{Outcome\ from\ cross-pollination} \qquad (1)$$

$$AF = \frac{Outcome\ from\ pollinator\ exclusion}{Outcome\ from\ cross-pollination} \qquad (2)$$

AF, thus, did not exclude apomixis, which is an asexual mode of reproduction by seed. Each index was calculated for fruit set and for seed production per flower.

An index of zero indicates a low degree of SC or AF; and a value of one indicates a high degree of SC or AF[22,35]. Values larger than one occurred when the outcome from self-pollination or pollinator exclusion was larger than that from cross-pollination[22]. We set such values larger than one to one (values set to one were up to 16% of observations; also see ref. 25). We averaged the indices for each

species within a study. This means that if a study reports selfing-ability data for a species in multiple locations, we used an averaged index. However, if a species was reported in more than one study, we included multiple indices for this species. The maximum number of indices for a species was three (Supplementary Data 1).

To assess potential relationships between SC and AF on global naturalization, we used two approaches. First, we used a generalized linear model approach that allowed us to correct for phylogenetic non-independence of species by including a variance-covariance matrix that contains the phylogenetic distance among species combined with a model of evolution. Second, we used path analysis, in which we could not account for phylogeny, but which allowed us to unravel direct and indirect effects of SC and AF on naturalization.

To test whether naturalization is associated with the degree of SC and AF of species, respectively, we fitted a phylogenetic logistic regression[63] using the package 'phylolm'[64] in R. As a response variable, we used naturalization incidence, that is, whether a species has or has not been recorded as naturalized outside its native ranges, as indicated by its presence/absence in the GloNAF database. As predictor variables, we included native range size (natural log-transformed number of TDWG level-2 regions), annual/biennial life history (yes/no, no if perennial), index of SC or AF, and its interaction with life history. We had such data for a total of 1,115 species from 481 studies for the analysis of SC, and 828 species from 373 studies for that of AF, based on fruit set. As some species were reported in more than one study, we considered the unique combination species-study source as the unit of observation. Therefore, we used more than one index (maximum of three) of SC or AF for 5% and 4% of the species, respectively. Nevertheless, the results remained qualitatively similar when we used one average index per species (Supplementary Table 4). To facilitate comparisons among estimates, we standardized the covariates to a mean of zero and an s.d. of 1 (ref. 65). To quantify the variance explained by the variables in our models, we reported the $R^2$ for the phylogenetic linear regressions[66], and we calculated the relative importance of each variable as the difference in deviance between the full model and a model without the variable of interest. Although $R^2$ can be calculated for non-phylogenetic logistic regression[66] (see values in Supplementary Table 1), such an approach quantifying the $R^2$ is not available for the phylogenetic logistic regressions.

To test whether the extent of naturalization for species naturalized in at least one region is positively related to its degree of SC and AF, respectively, we analysed the subset of species with breeding-system data that are listed in the GloNAF database. This subset comprised 274 species from 168 studies for SC indices, and 195 species from 114 studies for AF indices, based on fruit set. We fitted phylogenetic linear regressions[67] using the package 'phylolm' in R. As a response variable, we used naturalization extent measured as the natural log-transformed number of regions where the species is naturalized. As predictor variables, we used the same covariates as in the phylogenetic logistic regression described above.

We additionally analysed the complex association of SC and AF of plant species with their life history (annual/biennial or perennial), their native range size and their naturalization outside the native range using path analysis[68]. To specify the path structure, we assumed the following relationships: (i) naturalization outside the native range is driven by the degree of SC and AF of a species, respectively, its native range size and life history (annual/biennial vs perennial); (ii) native range size of a species is driven by its degree of SC or AF, and life history; and (iii) the degrees of SC or AF of a plant species are correlated with an annual or a biennial life history. This path analytical approach allowed us to quantify the potential direct and indirect effects of SC or AF of a plant species on its naturalization outside the native range.

As in the previous models, we separately analysed naturalization incidence (whether the species have been recorded as naturalized outside their native ranges), and for the subset of only the naturalized species, their naturalization extent (the number of regions where they are naturalized). For the analysis of the binary variable naturalization incidence, we used a diagonally weighted least square estimation method with robust standard errors. For the analysis of naturalization extent, we log-transformed the number of regions where the species is native to and the number of regions, where the species is naturalized, and used a maximum likelihood estimation method with robust standard errors (MLR). To facilitate comparisons among the different path coefficients, we standardized all continuous variables to a mean of zero and s.d. of 1. We assessed model fit using a relative measure, the Comparative Fit Index, and an absolute measure, the root mean square error of approximation. A Comparative Fit Index >0.95 and a root mean square error of approximation <0.06 are considered good[69]. In addition, we reported $R^2$ for the endogenous variables naturalization incidence and naturalization extent. We did the path analysis using the 'lavaan' package[70] in R. The results remained qualitatively similar when we used one average index per species (Supplementary Fig. 5).

**Data availability.** Data for this article, including four selfing-ability indices and information on data sources, species name, life history, native range and global naturalization as assessed based on the Global Naturalized Alien Flora database may be found in the Supplementary Information (Supplementary Data 1). Fruit set and seed production data in pollinator exclusion, self-pollination and cross-pollination treatments are available from the corresponding author upon request.

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

## Acknowledgements

We thank A. Raduski and collaborators for sharing their database on self-incompatibility; E. Rehn for acquiring PDF files of many studies; B. Hodapp and D. Wagner for their help in data extraction; the International Max Planck Research School for Organismal Biology and the Equal Opportunity Council of the University of Konstanz for supporting M.R.; and the DFG (Projects KL1866/3-1 and KL1866/9-1) for funding. P.P. and J.P. were supported by long-term research development project RVO 67985939 (The Czech Academy of Sciences), 504/11/1028 (GA CR) and project no. 14-36079G, Centre of Excellence PLADIAS (Czech Science Foundation). P.P. appreciates the support by Praemium Academiae award from The Czech Academy of Sciences. H.K. and P.W. acknowledge funding in the scope of the BEFmate project from the Ministry of Science and Culture of Lower Saxony. M.W. acknowledges the support of the German Centre for Integrative Biodiversity Research (iDiv) Halle-Jena-Leipzig, funded by the German Research Foundation (FZT 118).

## Author contributions

M.R. and M.v.K. compiled the global breeding-system database. W.D., F.E., H.K., J.P., P.P., P.W. and M.v.K. compiled the Global Naturalized Alien Flora database. M.R. and M.v.K. developed the research questions and analysed the data. N.M. contributed to the data analyses. M.R. wrote the manuscript with input from M.v.K., N.M., W.D., F.E., H.K., J.P., P.P., P.W. and M.W.

## Additional information

**Competing financial interests:** The authors declare no competing financial interests.

