## [Peer Review File · Nature Communications]

Reviewers' comments:

Reviewer #1 (Remarks to the Author):

OVERALL COMMENTARY

This is an interesting and well-written manuscript, based on a wealth of data the authors have assembled through the literature, combining a global breeding-system database of 1,752 angiosperm species with the Global Naturalized Alien Flora (GloNAF) database of globally naturalized plants, and analyzed them using novel and appropriate statistical methods. Unfortunately none of these databases are available, but from the list appearing in the supplementary table 1 I can see that there are plant species (e.g. invasives with published seed/fruit set) missing (see below).

After having read the manuscript carefully, I find it worth publishing, as it gives a very comprehensive picture of the importance of mating system and life habit on plant naturalization worldwide. I have only minor points to make (see below).

SPECIFIC COMMENTS

- At least a fraction, if not the entire, of the database first reported here (viz. global breeding-system database) should be included as part of the supplementary material.
- There are many cases in the literature for which there are more than one values of autofertility or self-compatibility measured in different areas of the world. Which values were used in the manuscript?
- Line 41: replace "and the Anthropocene" with "in the Anthropocene".
- Lines 70-72: a reference is needed.

Table 1

- Make clear that the first set of results (between entire lines) refers to self-compatibility, the second to autofertility.
- Line 607: "variables were centered and scaled": this needs explanation.

Supplementary table 1

- What about species included in the list without any number, which means without indices? For some species, however, there are indices reported in the literature (e.g. *Solanum elaeagnifolium* and others) but they are not included in this list.

Reviewer #2 (Remarks to the Author):

This manuscript presents the results of a study aimed at testing the hypothesis that the ability of a plant to self-fertilize should affect its tendency to naturalize in areas outside its native range. The hypothesis is important, not only because understanding the plant traits associated with naturalisation potential remains a crucial task in our quest to comprehend invasiveness (given that naturalisation is a first step in the invasion process), but also because there is on-going discussion and debate over the role of a selfing ability in the colonisation process, i.e., in the basis of 'Baker's Law' (which suggests that long-distance colonisation should be facilitated by a capacity for uniparental reproduction, in particular an ability to self-fertilize. While there have been a number of studies addressing the central question posed by the authors of this manuscript, some supporting the prediction and others not, the novelty of this study is principally in the global and phylogenetic scope of the data analysed, as well as the explicit inclusion of other potentially direct and indirect effects of plant traits on the naturalisation success, notably the size of the native range and the species' life history. The establishment of a database linking mating-system estimates with variables relevant to naturalisation and ultimately invasiveness (within a phylogenetic context) represents in itself an important contribution.

This article takes an important step forward in tests of Baker's Law, and it will be of interest to a relatively large readership. The database is also likely to be useful to researchers working on mating-system and life-history evolution. I am not familiar with all the details of the statistical methods employed in the study, but I am sufficiently satisfied that the approach taken rests on solid ground. The paper is well written, concise, and fair and measured in its interpretations and conclusions, and indeed is a pleasure to read. I have only one substantive comment, that I think is critical to the likely impact of this paper. This concerns the effect sizes of the traits and variables tested.

Although correlation coefficients are given in the results, the text makes rather vague claims about how important the effects in fact are. We are told, for example, that "species widespread at home tend to be more likely to naturalize" (line 218); "Annual and biennial species were more likely to be naturalised elsewhere" (line 229); "along with selfing ability, annual and biennial life forms contribute to alien plant naturalization" (line 239); and "selfing ability plants a significant role in the naturalization of alien plant species" (line 241). But how large are these effects? A strength of the study is in the size of the database assembled, but large sample sizes of course mean that very small effect sizes can end up being significant. The importance of this paper thus depends on how much a selfing ability increases the probability of naturalization or the range size of a naturalized species. In the absence of clear statements about the effect sizes, we are left somewhat in the dark about how important these results really are.

Reviewer #1

OVERALL COMMENTARY

This is an interesting and well-written manuscript, based on a wealth of data the authors have assembled through the literature, combining a global breeding-system database of 1,752 angiosperm species with the Global Naturalized Alien Flora (GloNAF) database of globally naturalized plants, and analyzed them using novel and appropriate statistical methods. Unfortunately none of these databases are available, but from the list appearing in the supplementary table 1 I can see that there are plant species (e.g. invasives with published seed/fruit set) missing (see below). This is the reason I believe the database, at least the “global breeding-system” one should become available which certainly will push up the readability of the paper.

RESPONSE: With the revised version of our manuscript, we provide the data from the breeding system database in an Excel file with information on species, life form, four selfing ability indices, native range, global naturalization as assessed based on GloNAF database, and data source (Supplementary Data 1). Regarding the information provided in Supplementary Table 1 in the previous version of the manuscript, we did a rigorous check of our database, which allowed us to confirm that we could not calculate the selfing ability indices for all species due to missing data, infinite values (the denominator i.e. the outcome of cross-pollination is zero) and undefined values (both numerator and denominator are zero), except for three species that were missing. The three species for which selfing ability data could be calculated using seed production per flower, but were mistakenly omitted from the data were *Centaurea solstitialis*, *Echium plantagineum* and *Solanum elaeagnifolium*. We now included those indices in the data, and redid the analysis. The results of the phylogenetic logistic and linear regressions remain qualitatively similar in direction, magnitude and significance (Supplementary Table 2). However, in the path analysis, the marginally significant positive effects of self-compatibility on naturalization extent and on native range size became significant (Supplementary Fig. 7).

After having read the manuscript carefully, I find it worth publishing, as it gives a very comprehensive picture of the importance of mating system and life habit on plant naturalization worldwide. I have only minor points to make (see below).

SPECIFIC COMMENTS

- At least a fraction, if not the entire, of the database first reported here (viz. global breeding-system database) should be included as part of the supplementary material.

RESPONSE: As we mentioned above, we now provide the data in an Excel file (Supplementary Data 1).

- There are many cases in the literature for which there are more than one values of autofertility or self-compatibility measured in different areas of the world. Which values were used in the manuscript?

RESPONSE: We averaged the indices for each species within a study. This means that if a study reports selfing ability data for a species in multiple locations, we used an averaged index. However, if a species was reported in more than one study, we included multiple indices for this species. We also did the analyses with only a single value per species (i.e. averaged across different studies), which gave similar results (see Supplementary Table 5 and Supplementary Fig. 9). Nevertheless, our analyses might not capture all information on the intraspecific variation of selfing ability, so future studies should test for a large number of species whether intraspecific variation in selfing ability plays a role in species naturalization. We now explain this more clearly in the Methods (lines 389-393). In addition, we now discuss the intraspecific variation as follows: ‘Furthermore, selfing ability can vary within species (e.g. in different locations⁴⁶). The roles of intraspecific variation in selfing ability and potential short- and long-term disadvantages of selfing for naturalized alien plants require further tests’ (lines 217-219).

- Line 41: replace "and the Anthropocene" with "in the Anthropocene".

RESPONSE: We replaced "and the Anthropocene" with "in the Anthropocene" (line 41).

- Lines 70-72: a reference is needed.

RESPONSE: We now cite a study by Mena-Ali et al. 2012 on partial self-incompatibility (line 73).

Table 1

- Make clear that the first set of results (between entire lines) refers to self-compatibility, the second to autofertility.

RESPONSE: We added ‘Analysis with self-compatibility’ and ‘Analysis with autofertility’ to clarify this (Table 1).

- Line 607: "variables were centered and scaled": this needs explanation.

RESPONSE: We now reformulated this as: ‘Variables were rescaled to have a mean of zero and a standard deviation of one’ (line 672). In the Methods, we explain why this was done (lines 413-415).

Supplementary table 1

- What about species included in the list without any number, which means without indices? For some species, however, there are indices reported in the literature (e.g. *Solanum elaeagnifolium* and others) but they are not included in this list.

RESPONSE: As we mentioned above, we could not calculate selfing ability indices for all (except three missing) species due to missing data, infinite values and undefined values. Three species (*Centaurea solstitialis*, *Echium plantagineum* and *Solanum elaeagnifolium*) that were mistakenly omitted from the dataset are now included in the analysis. The results of the phylogenetic logistic and linear regressions did not qualitatively change, and some of the effects in the path analysis are now stronger. We now replaced the former Supplementary Table 1 by an Excel file with the data (Supplementary Data 1).

Reviewer #2

This manuscript presents the results of a study aimed at testing the hypothesis that the ability of a plant to self-fertilize should affect its tendency to naturalize in areas outside its native range. The

hypothesis is important, not only because understanding the plant traits associated with naturalisation potential remains a crucial task in our quest to comprehend invasiveness (given that naturalisation is a first step in the invasion process), but also because there is on-going discussion and debate over the role of a selfing ability in the colonisation process, i.e., in the basis of 'Baker's Law' (which suggests that long-distance colonisation should be facilitated by a capacity for uniparental reproduction, in particular an ability to self-fertilize). While there have been a number of studies addressing the central question posed by the authors of this manuscript, some supporting the prediction and others not, the novelty of this study is principally in the global and phylogenetic scope of the data analysed, as well as the explicit inclusion of other potentially direct and indirect effects of plant traits on the naturalisation success, notably the size of the native range and the species' life history. The establishment of a database linking mating-system estimates with variables relevant to naturalisation and ultimately invasiveness (within a phylogenetic context) represents in itself an important contribution.

This article takes an important step forward in tests of Baker's Law, and it will be of interest to a relatively large readership. The database is also likely to be useful to researchers working on mating-system and life-history evolution. I am not familiar with all the details of the statistical methods employed in the study, but I am sufficiently satisfied that the approach taken rests on solid ground. The paper is well written, concise, and fair and measured in its interpretations and conclusions, and indeed is a pleasure to read. I have only one substantive comment, that I think is critical to the likely impact of this paper. This concerns the effect sizes of the traits and variables tested.

RESPONSE: In this new version, to quantify the explained variance by the variables in our models, we consistently report R^2 for the phylogenetic linear regressions following the approach described by Nakagawa and Schielzeth in *Methods in Ecology and Evolution* (2013). Additionally, we calculated the relative importance of each variable as the difference in deviance between the full model and a model without the variable of interest following the approach by Kempel et al. in *PNAS* (2013). Unfortunately, an approach to calculate R^2 is not available for the phylogenetic logistic regressions (Anthony Ives, personal communication). We mention this in the manuscript as follows: 'To quantify the variance explained by the variables in our models,

we reported the R^2 for the phylogenetic linear regressions⁶⁶, and we calculated the relative importance of each variable as the difference in deviance between the full model and a model without the variable of interest. Although R^2 can be calculated for non-phylogenetic logistic regression⁶⁶ (see values in Supplementary Table 1), such an approach quantifying the R^2 is not available for the phylogenetic logistic regressions (A. Ives, pers. comm.)' (lines 415-420). In the path analysis, we reported the R^2 for the endogenous variables naturalization incidence and naturalization extent. We mention this as follows: 'Additionally, we reported R^2 for the endogenous variables naturalization incidence and naturalization extent' (lines 453-454).

Although correlation coefficients are given in the results, the text makes rather vague claims about how important the effects in fact are. We are told, for example, that "species widespread at home tend to be more likely to naturalize" (line 218); "Annual and biennial species were more likely to be naturalised elsewhere" (line 229); "along with selfing ability, annual and biennial life forms contribute to alien plant naturalization" (line 239); and "selfing ability plants a significant role in the naturalization of alien plant species" (line 241). But how large are these effects? A strength of the study is in the size of the database assembled, but large sample sizes of course mean that very small effect sizes can end up being significant. The importance of this paper thus depends on how much a selfing ability increases the probability of naturalization or the range size of a naturalized species. In the absence of clear statements about the effect sizes, we are left somewhat in the dark about how important these results really are.

RESPONSE: To provide clear statements about the effect sizes of our results and allow for assessing their importance as suggested by the referee, we now improved the description of our results. Because we standardized all our variables, both in the logistic and linear regressions and in the path analysis, one can interpret them in terms of effect sizes. We now write 'as an increase of one standard deviation (0.854 in the analysis with self-compatibility, and 0.848 in the analysis with autofertility) in the log-transformed number of regions where a species is native increased its odds of being naturalized somewhere by a factor of $e^{0.710} = 2.034$ and $e^{0.720} = 2.054$ (Table 1)' (lines 128-131). 'Moreover, annuals and biennials had a higher naturalization incidence than perennials, as an annual or a biennial life form increased a species' odds of being naturalized somewhere by a factor of $e^{0.731} = 2.077$ compared to a perennial life form, at least so in the analysis with autofertility (Table 1)' (lines 131-135). 'An increase of one standard deviation in

the self-compatibility index (0.410) and autofertility index (0.374) of a species increased the log-transformed number of regions where it is naturalized by 0.256 and 0.472, respectively (Table 1)' (lines 154-157). 'An annual or a biennial life form increased the log-transformed number of regions where a species is naturalized by 1.463 and 1.048 compared to a perennial life form in the analysis of self-compatibility and autofertility, respectively (Table 1)' (lines 159-161).

Additionally, we discussed the strength of these effects as follows: 'Our models explained 13% to 24% of the variation in naturalization incidence and extent of species (Table 1, Fig. 1), which is comparable with the explanatory power of models in previous ecological studies testing the importance of different traits on the naturalization or invasion of alien species (e.g. 36)' (lines 181-184). 'In contrast, selfing ability had a direct positive effect on the number of regions where a species has naturalized, and selfing ability significantly accounted for 36% to 59% of the explained variation in naturalization relative to the other variables in the models (Table 1)' (lines 188-190).

ADDITIONAL REMARK: As we noticed some mistakes in the branch length in our phylogenetic tree, we reran all the phylogenetic analyses using an updated tree with corrected branch lengths. The results of the new analyses remain qualitatively similar to the ones presented in the previous version of the manuscript, so our conclusions remain unchanged.

Yours sincerely,

Mialy Razanajatovo, Noëlie Maurel, Wayne Dawson, Franz Essl, Holger Kreft, Jan Pergl, Petr Pyšek, Patrick Weigelt, Marten Winter and Mark van Kleunen

REVIEWERS' COMMENTS:

Reviewer #1 (Remarks to the Author):

The authors have done an excellent job not only in putting together a wealth of literature data and carrying out a meta-analysis at a very large scale, but also in revising their manuscript which now reads well and answers to several hot questions including the one regarding the general validity of the Baker's law. I find that my queries and comments were satisfactorily addressed, but I still have a few comment to make:

1. Is it co-incidence that the plant list of the previous version amounted to 1,752 species, exactly the same number as this version which though includes some additional species? (supplementary data 1 contains, indeed, 1,752 species).
2. Lines 34-35: instead of "has naturalized" I would write "naturalizes"
3. Line 37: instead of "has naturalized" I would write "is naturalized"
4. Supplementary Data 1 (also Line 334): besides native range size, please add a column denoting whether the plant species occurs also as alien/invasive somewhere.

REVIEWERS' COMMENTS:

Reviewer #1 (Remarks to the Author):

The authors have done an excellent job not only in putting together a wealth of literature data and carrying out a meta-analysis at a very large scale, but also in revising their manuscript which now reads well and answers to several hot questions including the one regarding the general validity of the Baker's law. I find that my queries and comments were satisfactorily addressed, but I still have a few comment to make:

1. Is it co-incidence that the plant list of the previous version amounted to 1,752 species, exactly the same number as this version which though includes some additional species? (supplementary data 1 contains, indeed, 1,752 species).

RESPONSE:

Please note that the three additional species *Centaurea solstitialis*, *Echium plantagineum* and *Solanum elaeagnifolium* were already listed in the plant list of the previous version, but only the selfing ability indices were missing. In the previous version, we did not get selfing ability indices for these species because the breeding system data was mistakenly entered as number of seeds per fruit instead of as number of seeds per flower. The latter was used to calculate selfing ability indices based on seed production.

2. Lines 34-35: instead of "has naturalized" I would write "naturalizes"

RESPONSE:

After addressing the editor's comment, the updated statement is as follows "Selfing ability is associated with (bi)annual life history and a large native range, which both positively correlate with the probability of naturalization. Path analysis suggests that a high selfing ability directly increases the number of regions where a species is naturalized." (lines 34-37)

3. Line 37: instead of "has naturalized" I would write "is naturalized"

RESPONSE:

We edited as requested (line 37).

4. Supplementary Data 1 (also Line 334): besides native range size, please add a column denoting whether the plant species occurs also as alien/invasive somewhere.

RESPONSE:

This information is already provided with the variable 'Listed in GloNAF' in the Supplementary Data 1, which takes a value of 1 when a species has been reported as (naturalized) alien in one or more regions, and a value of 0 otherwise.